# Therapeutic Cell Protective Role of Histochrome under Oxidative Stress in Human Cardiac Progenitor Cells

**DOI:** 10.3390/md17060368

**Published:** 2019-06-21

**Authors:** Ji Hye Park, Na-Kyung Lee, Hye Ji Lim, Sinthia Mazumder, Vinoth Kumar Rethineswaran, Yeon-Ju Kim, Woong Bi Jang, Seung Taek Ji, Songhwa Kang, Da Yeon Kim, Le Thi Hong Van, Ly Thanh Truong Giang, Dong Hwan Kim, Jong Seong Ha, Jisoo Yun, Hyungtae Kim, Jin Han, Natalia P. Mishchenko, Sergey A. Fedoreyev, Elena A. Vasileva, Sang Mo Kwon, Sang Hong Baek

**Affiliations:** 1Laboratory of Regenerative Medicine and Stem Cell Biology, Department of Physiology, Medical Research Institute, School of Medicine, Pusan National University, Yangsan 50612, Korea; siwonvin@naver.com (J.H.P.); ahlng2005@naver.com (N.-K.L.); dla9612@naver.com (H.J.L.); sinthiambbs@gmail.com (S.M.); vinrebha@gmail.com (V.K.R.); twou1234@nate.com (Y.-J.K.); jangwoongbi@naver.com (W.B.J.); jst5396@hanmail.net (S.T.J.); songhwa.kang@gmail.com (S.K.); ekdus0258@gmail.com (D.Y.K.); lethihongvan25121978@gmail.com (L.T.H.V.); lythanhtruonggiang@gmail.com (L.T.T.G.); jongseong@pusan.ac.kr (J.S.H.); jsyun14@hanmail.net (J.Y.); 2Research Institute of Convergence Biomedical Science and Technology, Pusan National University School of Medicine, Yangsan 50612, Korea; 3Department of Neurosurgery & Medical Research Institute, Pusan National University Hospital, Busan 49241, Korea; smile0402@hanmail.net; 4Department of Thoracic and Cardiovascular Surgery, Pusan National University Yangsan Hospital, Yangsan 50612, Korea; 2719k@naver.com; 5National Research Laboratory for Mitochondrial Signaling, Department of Physiology, Department of Health Sciences and Technology, BK21 Plus Project Team, Cardiovascular and Metabolic Disease Center, Inje University College of Medicine, Busan 47392, Korea; phyhanj@gmail.com; 6G.B. Elyakov Pacific Institute of Bioorganic Chemistry, Far-Eastern Branch of the Russian Academy of Science, Vladivostok 690022, Russia; mischenkonp@mail.ru (N.P.M.); fedoreev-s@mail.ru (S.A.F.); vasilieva_el_an@mail.ru (E.A.V.); 7Division of Cardiology, Seoul St. Mary’s Hospital, School of Medicine, the Catholic University of Korea, Seoul 06591, Korea

**Keywords:** cardiac progenitor cells, histochrome, echinochrome A, oxidative stress, cell therapy

## Abstract

Cardiac progenitor cells (CPCs) are resident stem cells present in a small portion of ischemic hearts and function in repairing the damaged heart tissue. Intense oxidative stress impairs cell metabolism thereby decreasing cell viability. Protecting CPCs from undergoing cellular apoptosis during oxidative stress is crucial in optimizing CPC-based therapy. Histochrome (sodium salt of echinochrome A—a common sea urchin pigment) is an antioxidant drug that has been clinically used as a pharmacologic agent for ischemia/reperfusion injury in Russia. However, the mechanistic effect of histochrome on CPCs has never been reported. We investigated the protective effect of histochrome pretreatment on human CPCs (hCPCs) against hydrogen peroxide (H_2_O_2_)-induced oxidative stress. Annexin V/7-aminoactinomycin D (7-AAD) assay revealed that histochrome-treated CPCs showed significant protective effects against H_2_O_2_-induced cell death. The anti-apoptotic proteins B-cell lymphoma 2 (Bcl-2) and Bcl-xL were significantly upregulated, whereas the pro-apoptotic proteins BCL2-associated X (Bax), H_2_O_2_-induced cleaved caspase-3, and the DNA damage marker, phosphorylated histone (γH2A.X) foci, were significantly downregulated upon histochrome treatment of hCPCs in vitro. Further, prolonged incubation with histochrome alleviated the replicative cellular senescence of hCPCs. In conclusion, we report the protective effect of histochrome against oxidative stress and present the use of a potent and bio-safe cell priming agent as a potential therapeutic strategy in patient-derived hCPCs to treat heart disease.

## 1. Introduction

Stem cells have been reported to recover damaged hearts from myocardial infarction and have been investigated for use in myocardial regeneration [1,2,3]. Cardiac progenitor cells (CPCs) were first identified by Anversa et al. [4]. CPCs are classified as a prevailing stem cell population in the heart and have crucial roles in cardiac homeostasis [5,6]. CPCs can differentiate into multiple cell lineages of the heart, and thus, are a promising cell resource for regenerating ischemic hearts [7]. Recent preclinical studies suggest that transplantation of CPCs into the ischemic myocardium can significantly improve cardiac regeneration via the formation of vasculature and new myocytes [8,9,10]. Furthermore, CPCs have the potential to produce and remodel extracellular matrix (ECM) proteins [11], trigger CPC proliferation, and stimulate growth factor secretion [12]. According to these positive results, CPC might be a promising stem cell source in cardiovascular regeneration. However, current evidence suggests that poor viability of engrafted CPCs in the infarcted myocardium primarily restricts the therapeutic efficacy of CPCs [13,14]. Thus, increasing the survival of CPCs can be a beneficial strategy to enhance the therapeutic effect in ischemic heart disease.

Reactive oxygen species (ROS), such as hydrogen peroxide (H_2_O_2_), superoxide radicals, and hydroxyl radicals, are produced during infarction or reperfusion of ischemic hearts [15]. ROS involvement has been reported in various important development processes, cell signaling, and regulation of homeostasis [16]. Low levels of ROS are involved in the regulation of stem cell fate decision, stem cell proliferation, differentiation, and survival [17]. However, excessive ROS production leads to impaired cell metabolism and decreased cell viability [18], thereby inhibiting transplanted CPCs to regenerate the damaged heart [19]. Consequently, protecting CPCs from undergoing apoptosis and enhancing their ability to survive under oxidative stress is crucial in optimizing CPC-based therapy.

Echinochrome A is a common sea urchin pigment [20] that has a chemical structure of 6-ethyl-2,3,5,7,8-pentahydroxy-1,4-naphthoquinone (Figure 1A) and exhibits antioxidant, anti-viral [21], anti-inflammatory [22], and anti-diabetic activities [23]. Echinochrome A prevents mitochondrial dysfunction and activation of mitogen-activated protein kinase (MAPK) cell death signaling pathways caused by cardiotoxic drug treatment [24]. Echinochrome A regulates mitochondrial biogenesis in cardiomyocytes by upregulating the transcription of mitochondrial regulatory genes, such as mitochondrial transcriptional factor A (TFAM), nuclear respiratory factor (NRF-1), and proliferator-activated receptor gamma co-activator (PGC-1α) [25]. Echinochrome A inhibits the phosphorylation of serine-16 and threonine-17, located in the active center of phospholamban (membrane phosphoprotein and main regulator of the SERCA2A receptor responsible for the transfer of calcium ions from the cytosol to the sarcoplasmic reticulum), preventing ischemic myocardial damage by reducing the infarction zone [26].

Echinochrome A is insoluble in water, however, its water-soluble sodium salt is used for medical applications, which is manufactured under inert conditions in ampoules and is known as the Histochrome^®^ drug. Histochrome has been used in Russia in ophtalmological and cardiological clinical practice. In ophthalmology, histochrome is used for the treatment of degenerative diseases of the retina and cornea, macular degeneration, primary open-angle glaucoma, diabetic retinopathy, hemorrhage in the vitreous body, retina, and anterior chamber, and dyscirculatory disorder in the central artery and vein of the retina [27]. An overview of clinical applications of histochrome in cardiology is presented in monography [28]. In the first place, histochrome has been used for the treatment of myocardial ischemia/reperfusion injury. Even a single injection of histochrome immediately after reperfusion recovered the ECG signs of myocardial necrosis and significantly (up to 30%) reduces the necrosis zone after a 10-day course. The use of histochrome prevented lipid peroxidation, reduced the frequency of left ventricular failure, did not affect the level of blood pressure and heart rate, and decreased the frequency of post-infarction angina pectoris. Practical experience of histochrome treatment confirmed the absence of any adverse effects and the safety of its application [28].

The cardioprotective effect of histochrome on patient-derived CPCs has never been reported. Thus, we investigated whether pretreatment of CPCs with histochrome promotes cell survival against oxidative stress during cardiac regeneration.

## 2. Results

### 2.1. Histochrome Does Not Affect Surface Expression Markers of Human Cardiac Progenitor Cells (hCPCs)

To evaluate the cytotoxicity of histochrome in human CPCs (hCPCs), hCPCs were treated with different concentrations of histochrome for 24 h. Cell survival was found to be significantly increased for 0.5 μM to 10 μM of histochrome and significantly decreased at concentrations above 100 μM (*p* < 0.01 versus 0 μM; Figure 1B). Based on the data obtained, we determined that histochrome concentration under 50 μM used for the further experiments. No change in the morphology of hCPCs was observed on pretreatment with 0 μM, 5 μM, 10 μM, and 20 μM concentrations of histochrome (Figure 1C). To eliminate the possibility of change in CPC characteristics on pretreatment with histochrome, we investigated typical surface expression markers of hCPCs using fluorescence-activated cell sorting (FACS) analysis. As shown in Figure 1D, histochrome-treated CPCs showed positive expression of cardiac stem cell markers such as mast/stem cell growth factor receptor kit (c-kit), cluster of differentiation 66 (CD166), CD29, CD105, and CD44. However, negative expression was observed for hematopoietic markers, such as CD45 and CD34, in pretreated hCPCs compared to that in control cells.

### 2.2. Histochrome Reduced Cellular and Mitochondrial Reactive Oxygen Species (ROS) Levels in hCPCs during H_2_O_2_-Induced Oxidative Stress

To investigate whether pretreating hCPCs with histochrome protects them against oxidative stress, we performed a cellular ROS staining assay. Cellular ROS-tagged green intensity was found to be significantly increased upon exposure to H_2_O_2_ (Figure 2A). We observed that pretreatment with histochrome decreased the cellular ROS levels in a dose-dependent manner. The 2’,7’–difluorofluorescin diacetate (H_2_-DFFDA) assay revealed that pretreatment with 10 μM of histochrome significantly decreased cellular ROS levels (Figure 2B). Furthermore, we investigated the effects of pretreatment with histochrome on mitochondrial superoxide production in hCPCs. The increased production of mitochondrial superoxide caused by H_2_O_2_ addition was found to be significantly reduced in histochrome-treated hCPCs (Figure 2C). Our data suggested that histochrome has intracellular ROS scavenging activity in hCPCs under oxidative stress.

### 2.3. Anti-Apoptotic Effect of Histochrome against H_2_O_2_-Induced Cell Death

To investigate the anti-apoptotic effects of histochrome in hCPCs, cells were treated with 1 mM H_2_O_2_ for 4 h and cell apoptosis was evaluated using flow cytometry by staining with Annexin V and 7-aminoactinomycin D (7-AAD). Annexin V/7-AAD assay revealed that treatment with H_2_O_2_ significantly increased the percentage of apoptotic cells (+H_2_O_2_, -Histochrome; 14.3% ± 2.36%) compared to the percentage of apoptotic cells in the non-treated control group (-H_2_O_2_, -Histochrome; -8.7% ± 0.84%, -H_2_O_2_, +Histochrome; 5.6% ± 0.40%, Figure 3A). In contrast, pretreatment of hCPCs with 10 µM of histochrome significantly increased the percentage of viable cells (+H_2_O_2_, +Histochrome; 95.2% ± 0.40%, Figure 3A) while decreasing the number of apoptotic cells (+H_2_O_2_, +Histochrome; 2.4% ± 0.49%, Figure 3A). We also investigated the effect of histochrome on cell morphology using phase contrast microscopy (Figure 3B). H_2_O_2_ treatment caused abnormal morphology and reduced cell viability. However, pretreatment with histochrome attenuated the morphological change induced by H_2_O_2_. In addition, live cell imaging analysis suggested that pretreatment with histochrome under the H_2_O_2_-induced oxidative stress condition significantly increased the number of live cells in a dose-dependent manner (Figure 3C).

### 2.4. Histochrome Protects hCPCs against Oxidative Stress through Downregulation of Pro-Apoptotic Signals and Upregulation of Anti-Apoptotic Signals

Further, we investigated the expression of apoptosis-related proteins by western blotting. Pretreatment with histochrome was observed to reduce expression of pro-apoptotic protein, Bcl-2 associated X (Bax) while promoting the expression of anti-apoptotic proteins, B-cell lymphoma 2 (Bcl-2), and Bcl-xL. In addition, pretreatment with histochrome remarkably decreased the expression levels of cytochrome C and cleaved-caspase-3 (Figure 4A). Our data suggested that histochrome protects hCPCs against oxidative stress through decreased pro-apoptotic signals and increased anti-apoptotic signals. Further, we checked DNA damage marker, phosphorylated histone (γH2A.X) foci by immunocytochemistry (Figure 4B). We found that pretreatment of hCPCs with histochrome prevented DNA damage caused by oxidative stress. Overall, our results suggested that pretreatment with histochrome regulates cell apoptosis by altering cell survival signals and inhibiting ROS-induced DNA damage.

### 2.5. Prolonged Treatment with Histochrome Attenuates Cellular Senescence in hCPCs

Several antioxidants have been reported to be associated with inhibition of cellular senescence [29,30,31]. To examine whether histochrome affects cellular senescence, we investigated the long-term effect of histochrome treatment on an in vitro culture of hCPCs. Evaluation of senescence-associated β galactosidase (SA-β-gal) activity in senescent hCPCs (passage 13) revealed a significant increase in the number of SA-β-gal positive cells (sene; 40.7% ± 3.07%) (Figure 5). However, increased SA-β-gal positive cells were significantly downregulated (sene+Histochrome; 31.3% ± 2.87%) on prolong treatment of hCPCs with histochrome. This suggested that long-term treatment of histochrome alleviates replicative cellular senescence in hCPCs.

## 3. Discussion

To enhance the engraftment rate of transplanted cells, several studies have been performed focusing on cell priming, using physical stimulations such as heat shock, and chemical stimulations using natural products and growth factors. Recently, stem cell priming using natural products has been proposed as a new strategy to enhance the cell activity and promote stable and efficient cell functioning. In our previous study, we demonstrated that pretreatment with fucoidan, a marine-sulfated polysaccharide derived from seaweeds, inhibits cellular senescence and promotes neo-vasculogenic potential [32].

In this study, we identified a novel priming factor for enhancing cell therapy potentials against oxidative stress. A recent study has revealed that echinochrome A promotes cardiomyocyte differentiation of mouse embryonic stem cells via direct binding to serine-threonine kinase PKCι and inhibition of its activity [33]. We are the first to report the effects of histochrome on hCPCs. In the present study, we demonstrated that pretreatment with histochrome enhanced the survival of hCPCs against oxidative stress. Further, pretreatment with different concentrations (0 µM, 5 µM, 10 µM, 15 µM, and 20 µM) of histochrome did not alter the expression of multipotent cardiac stem cell markers such as c-kit (Histo 0 μM; 99.9% ± 0.1%, Histo 5 μM; 99.9% ± 0.1%, Histo 10 μM; 99.8% ± 0.1%, and Histo 20 μM; 99.9% ± 0.1%, respectively); CD29 (Histo 0 μM; 99.9% ± 0.1%, Histo 5 μM; 99.6% ± 0.1%, Histo 10 μM; 99.1% ± 0.2%, and Histo 20 μM; 93.7% ± 5.6%, respectively); CD166 (Histo 0 μM; 88.9% ± 1.5%, Histo 5 μM; 80.2% ± 2.7%, Histo 10 μM; 80.9% ± 1.5%, and Histo 20 μM; 79.8% ± 2.1%, respectively); CD105 (Histo 0 μM; 95.0% ± 0.8%, Histo 5 μM; 98.5% ± 0.1%, Histo 10 μM; 97.8% ± 0.1%, and Histo 20 μM; 99.7% ± 0.1%, respectively); and CD44 (Histo 0 μM; 97.8% ± 1.0%, Histo 5 μM; 99.5% ± 0.1%, Histo 10 μM; 99.4% ± 0.2%, and Histo 20 μM; 99.6% ± 0.1%, respectively). In contrast, histochrome-treated hCPCs showed negative expression of hematopoietic markers such as CD34 (Histo 0 μM; 0.7% ± 0.01%, Histo 5 μM; 1.1% ± 0.1%, Histo 10 μM; 0.6% ± 0.1%, and Histo 20 μM; 1.0% ± 0.2%, respectively) and CD45 (Histo 0 μM; 0.1% ± 0.1%, Histo 5 μM; 0.1% ± 0.06%, Histo 10 μM; 0%, and Histo 20 μM; 0%, respectively).

In ischemic heart disease, ROS is produced upon reperfusion [34]. Excessive ROS formation promotes cell death which induces development and progression of cardiovascular disease [35]. The generation of ROS is associated with alteration in electrophysiology leading to myocardial ischemia [36]. In addition, mitochondrial ROS scavenging has been reported for its potential to prevent cardiac failure [37]. It is necessary to target intracellular ROS and mitochondrial ROS to prevent cell death caused by oxidative stress. Histochrome showed high intracellular ROS scavenging activity and reduced production of mitochondrial superoxide in hCPCs. Furthermore, pretreatment with histochrome revealed an anti-apoptotic effect against H_2_O_2_-induced cell death.

The Bcl-2-family members (Bcl-2, Bcl-xL, Bax) are predominant regulators of cell cycle progression and apoptosis [38]. Specifically, Bcl-2 and Bcl-xL are well-known negative regulators of apoptosis which promote cell survival [39]. On the other hand, the apoptosis regulator Bax inhibits cell survival [40] and cell cycle progression [41]. In this study, pretreatment with histochrome upregulated the expression of Bcl-2 and Bcl-xl and downregulated that of Bax under the H_2_O_2_-induced oxidative stress condition. DNA damage induces the formation of double stranded breaks (DSBs) and stimulates phosphorylation of histone H2AX [42]. Subsequently, γ-H2A.X is used as a reference marker for DNA damage. In our study, pretreatment of hCPCs with histochrome reduced the expression of the DNA damage marker, γH2A.X foci under the oxidative stress condition. Thus, we concluded that histochrome prevents ROS-mediated DNA damage in hCPCs (Figure 6).

Any cell undergoes a limited number of divisions and thus, following a certain number of cell cycles, it enters an irreversible cell cycle arrest which is referred to as cellular senescence [43]. Stem cell therapy for regeneration of tissues requires over hundreds of millions of cells [44]. Repetitive in vitro cultures are essential and cellular senescence is a major threat in such stem cell therapies. Thus, preventing cellular senescence is a promising strategy in stem cell therapy. SA-β-gal assay revealed that histochrome delayed the progression of cellular senescence in hCPCs. Thus, our study suggests that histochrome can be a potential effective strategy to overcome cellular senescence.

Overall, our present study demonstrated that histochrome protects hCPCs against oxidative stress by regulating cell survival signaling. Furthermore, histochrome prevents cellular senescence of hCPCs. Thus, our study presents a simple and effective strategy to improve cell survival in post-transplanted CPCs under ischemic oxidative stress conditions and improve the efficiency in myocardial regeneration.

## 4. Materials and Methods

### 4.1. Cell Cultures and Treatment

Human fetal right auricle(RA) tissue was received from Pusan National University YangSan Hospital and the study was approved by the Institutional Review Board (IRB) (IRB No. 05-2015-133). C-kit positive hCPCs were isolated and cultured, as previously reported [45,46]. hCPCs were maintained at 37 ℃ with 5% CO_2_ in Ham’s Nutrient Mixture F-12 (Hyclone, GE Healthcare, Chicago, IL, USA) and supplemented with 10% Fetal bovine serum (FBS; Gibco#16000-044, Thermo Fisher Scientific, Carlsbad, CA, USA), 1% penicillin-streptomycin (PS; Welgene, Daegu, Republic of Korea), 0.005 unit/mL of human erythropoietin (hEPO, R&D systems, Minneapolis, MN, USA), 10 ng/mL of recombinant human basic fibroblast growth factor (rb-FGF, Peprotech, Rocky Hill, NJ, USA), and 2 mM of glutathione (Sigma-Aldrich, St. Louis, CA, USA). Passages 4–10 were utilized for the experiments as passage 13 was found to be senescent cells. hCPCs were treated with different concentrations of H_2_O_2_ (Sigma-Aldrich, St. Louis, CA, USA) and a final concentration of 600 µM (to investigate the antioxidant effect), 1 mM (to confirm anti-apoptotic effect) was used.

The standardized substance echinochrome A (registration number in Russian Federation is P N002362/01) was isolated from the sea urchin *Scaphechinus mirabilis* by a previously described method [47]. The purity of echinochrome A (99.0%) was confirmed using LC-MS data (Shimadzu LCMS-2020, Kyoto, Japan). Purified echinochrome A appeared as red–brown needles, had a melting point of 221 °С, and similar nuclear magnetic resonance (NMR) spectra to that reported previously [47]. We used a solution of echinochrome A sodium salts in ampoules with trade name Histochrome^®^. Histochrome was generated by combining echinochrome A (1 g) with sodium carbonate (0.4 g) in a water solution heated in inert gas until CO_2_ was completely removed. This solution at a concentration of 0.2 mg/mL echinochrome A (750 μM) was sealed in ampoules in inert gas. After opening of the ampoule, histochrome was used as a stock solution to be diluted with appropriate solvents or culture media.

### 4.2. Cell Cytotoxicity Assay

To determine cell cytotoxicity, hCPCs were seeded at 5000 cells/well in a 96-well plate. The following day, cells were pretreated with different concentrations of histochrome and incubated for 24 h. Cell viability was evaluated using a cell-counting kit (CCK) cell viability assay kit (#CCK-3000, DonginLS, Seoul, Republic of Korea), following the manufacturer’s instructions [48,49]. The cell viability was measured by incubating cells with the CCK solution. The plates were incubated for 1 h, and the absorbance of each well was measured. Absorbance was measured at 450 nm using a microplate reader (TECAN, Mannedorf, Switzerland ). Cell viability of the experimental group was represented as percentage to 0 μM.

### 4.3. Western Blot Analysis

For western blotting, hCPCs pretreated with different concentrations of histochrome were exposed to 1 mM H_2_O_2_ (Sigma-Aldrich, St. Louis, CA, USA) for 4 hours. Further, cells were lysed in the Radioimmunoprecipitation assay (RIPA) buffer containing protease inhibitor and phosphatase inhibitor. Equivalent concentration of proteins (20 μg) were separated on sodium dodecyl sulphate-polyacrylamide gel electrophoresis (SDS-PAGE) and transferred to 0.45 μm PVDF membrane (Millipore, Billerica, MA, USA). The membranes were blocked with 5% skim milk and incubated with primary antibodies overnight at 4 ℃: cleaved-caspase 3 (Cell Signaling, Danvers, MA, USA), Bax, Bcl-2, Bcl-xL, cytochrome C, β-actin (Santa Cruz Biotechnology, Dallas, TX, USA), and γH2A.X (Abcam, Cambridge, UK). Further, the membranes were incubated with Horseradish peroxidase (HRP) -conjugated secondary antibody for 1 h at room temperature. The chemiluminescence signal were detected using enhanced chemiluminescence (ECL) reagent (Millipore, Billerica, MA, USA).

### 4.4. Immunocytochemistry

For immunocytochemistry, hCPCs were seeded at a density of 50,000 cells per well in a 2-well chamber slide (BioTek, Winooski, VT, USA). After pretreatment with histochrome for 24 h, cells were washed once with PBS and fixed in 4% paraformaldehyde (PFA) for 10 min. Cells were then permeabilized in 0.1% Triton X-100 + 0.01 M Glycine + Phosphate-buffered saline (PBS) solution for 30 min. Slides were blocked with 10% normal goat serum in PBST (0.1% Triton X-100 in PBS) and incubated for 1 h at room temperature. γH2A.X (phospho S139) antibody (#ab11174, 1:500 dilute in blocking buffer, Abcam, Cambridge, UK) was diluted in blocking solution and incubated overnight at 4 ℃. The following day, slides were washed 3 times with PBS and incubated with Alexa Flour 488 goat IgG anti-rabbit antibody (1:200, Invitrogen, Carlsbad, CA, USA), and incubated for 1 hour in the dark. Following cell wash twice, cells were mounted using ProLong diamond anti-fade mountant with 4′,6-diamidino-2-phenylindole (DAPI). Slides were analyzed under a Lionheart FX automated microscope (BioTek, Winooski, VT, USA)

### 4.5. Flow Cytometry Analysis

To examine c-kit expression, hCPCs were fixed in 4% PFA for 10 min and washed with PBS. The cells were fixed and permeabilized as previously described [50]. The fixed cells were incubated with c-kit antibody (#130-091-733, Miltenyi Biotec, Bergisch Gladbach, Germany) for 30 min at 4 ℃. For expression analysis of other stem cell markers, hCPCs were suspended in 100 μL FACS buffer (2 mM EDTA, 2% FBS in PBS solution) and incubated with CD34 (#55373, BD Bioscience, Franklin Lakes, NJ, USA), CD45 (#560839, BD Bioscience, Franklin Lakes, NJ, USA), CD166 (#559263, BD Bioscience, Franklin Lakes, NJ, USA), CD29 (#555443, BD Bioscience, Franklin Lakes, NJ, USA), CD105 (#560829, BD Bioscience, Franklin Lakes, NJ, USA), and CD90 (#130-097-930, Miltenyi Biotec, Bergisch Gladbach, Germany) FACS antibody. After incubation for 30 min, the cells were washed three times and resuspended in 100 μL FACS buffer. Expression of stem cell markers was analyzed on a BD Accuri flow cytometer (BD Bioscience, Franklin Lakes, NJ, USA).

### 4.6. ROS Measurement

Intracellular ROS levels were detected using CellROX green reagent (#C10444, Thermo Fisher Scientific, Carlsbad, CA, USA). Cells were stained according to the manufacturer’s instructions. Imaging and quantification of cellular ROS levels were performed using the CX7 High-Content Screening (HCS) System (Thermo Fisher Scientific, Carlsbad, CA, USA). For H_2_-DFFDA assay, hCPCs were harvested using Accutase (Sigma-Aldrich, St. Louis, CA, USA) and incubated with 10 μM carboxy-H_2_-DFFDA at 37 ℃ in the dark. After washing with PBS, the cells were resuspended in 100 μL of FACS buffer. The intracellular ROS level was assessed based on the fluorescence intensity of the cells.

### 4.7. Mitochondrial Superoxide Measurement

hCPCs were plated on a confocal plate (50,000 cells per well) and incubated overnight. The following day, cells were pretreated with/without histochrome (in 2% Ham’s F-12 medium) and incubated for 24 h. Cells were washed and incubated in low serum Ham’s F-12 medium supplemented with 5 μM MitoSOX Red (#M36008, Thermo Fisher Scientific, Carlsbad, CA, USA ) and 100 nM Mitotracker green (#M7514, Thermo Fisher Scientific, Carlsbad, CA, USA) for 30 min. Further, the cells were washed twice with PBS prior to analysis using a Lionheart FX automated microscope (BioTek, Winooski, VT, USA).

### 4.8. Cell Death Assay

hCPCs were pretreated with histochrome for 24 h in low serum (2% FBS) Ham’s F-12 medium. After incubation, cells were washed with PBS and exposed to 1 mM H_2_O_2_ for 4 h in low serum medium. Further, cells were harvested and suspended in 1 × annexin binding buffer supplemented with Annexin V (AnV; BD Pharminogen, #550475, San Diego, CA, USA) and 7-AAD(BD Pharminogen, San Diego, CA, USA).

### 4.9. Live Cell Imaging

For quantification of live cells in H_2_O_2_-induced oxidative stress, cells were stained with calcein (#R37601, Thermo Fisher Scientific, Carlsbad, CA, USA). Nuclei were stained with NucBlue Live Ready Probes Reagent (#R37605, Thermo Fisher Scientific, Carlsbad, CA, USA). Live cells were observed under the CX7 High-Content Screening (HCS) System (100×, Thermo Fisher Scientific, Carlsbad, CA, USA). Images were captured (25 images per well) and experiments were repeated four times. Green fluorescence indicated live cells.

### 4.10. Senescence Associated β-galactosidase Staining

Senescent cells were quantified by SA-β-gal assay kit (#9860, Cell Signaling Technology, Beverly, MA, USA) according to the manufacturer’s instructions. After X-gal staining, images were captured using an Olympus microscope (OLYMPUS, Tokyo, Japan). Each experiment was repeated three times.

### 4.11. Statistical Analysis

Data are presented as means ± standard deviation (SD). Statistical significance was assessed by student’s *t*-test to compare between the two groups. *p* < 0.05 was considered statically significant.

## 5. Conclusions

Pretreatment with histochrome (sodium salt of echinochrome A—a common sea urchin pigment) enhances cell survival under oxidative stress conditions by regulating the apoptosis signaling pathway and preventing DNA damage. Furthermore, histochrome attenuates cellular senescence in hCPCs. Our results suggest that pretreating c-kit-positive hCPCs with histochrome before transplantation might be a potential therapeutic strategy in treating ischemic heart disease.

## Figures and Tables

**Figure 1 marinedrugs-17-00368-f001:**
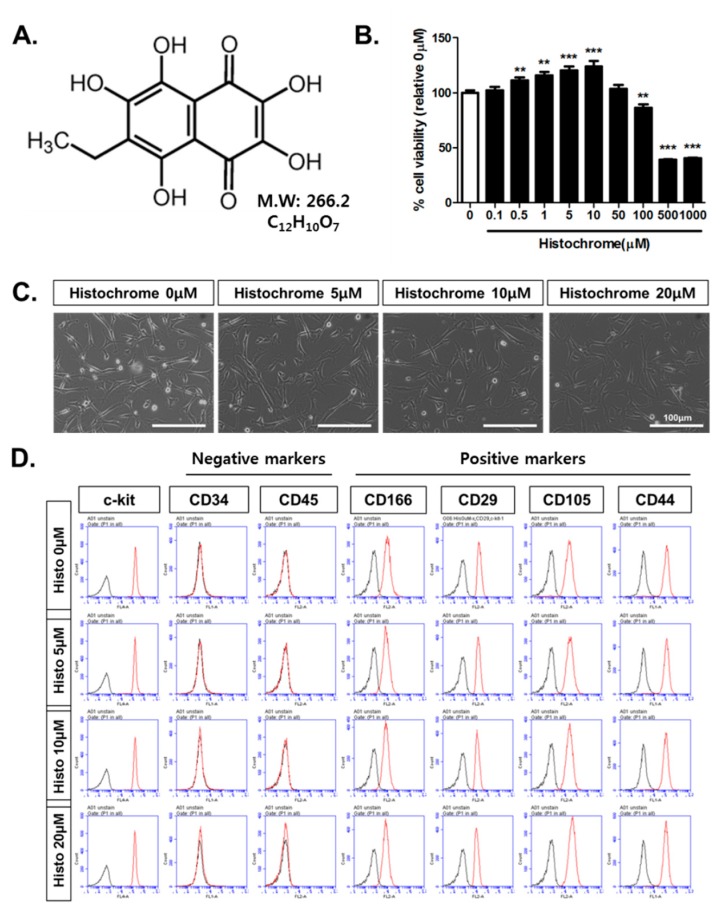
Effects of histochrome treatment on human cardiac progenitor cells (hCPCs) characterization. (**A**) Chemical structure of echinochrome A—active substance of the histochrome drug. (**B**) hCPCs were treated with different concentrations of histochrome for 24 h and viability was measured using cell viability, Proliferation & Cytotoxicity assay (CCK assay). Data are presented as the mean ± standard deviation (SD). **, *p* < 0.01 versus 0 μM, ***, *p* < 0.001 versus 0 μM. *n* = 6 (**C**) Morphological analysis of hCPCs pretreated with histochrome. Scale bar = 100 μm, (**D**) Expression of stem cell marker by flow cytometric analysis, *n* = 3. Error bars indicate standard effort of the mean (S.E.M)

**Figure 2 marinedrugs-17-00368-f002:**
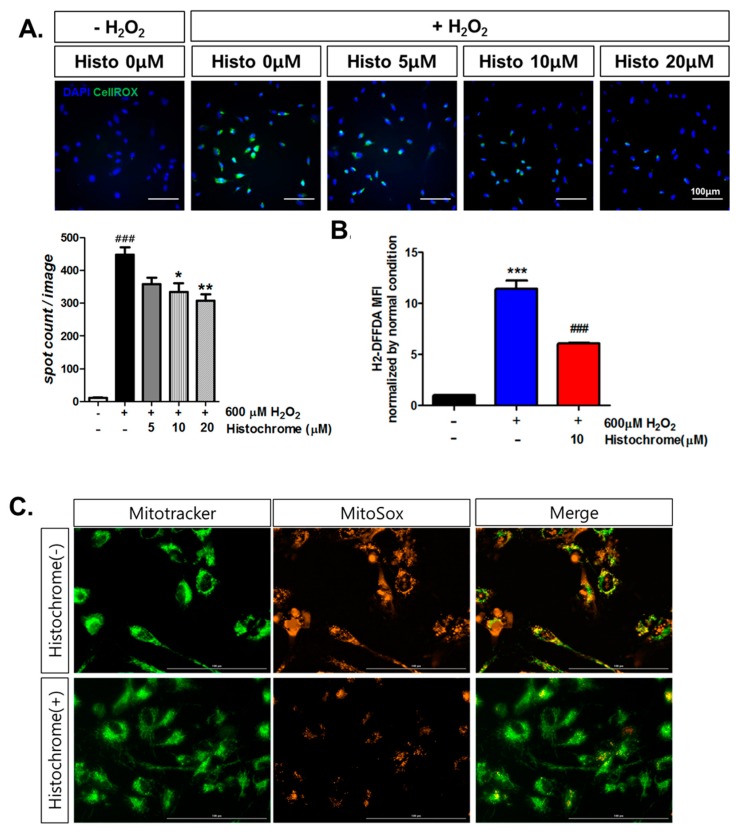
Intracellular reactive oxygen species (ROS) and mitochondrial ROS scavenging activity of histochrome in hCPCs. (**A**) hCPCs were pretreated with histochrome at 0 μM, 5 μM, 10 μM, and 20 μM for 24 h followed by the addition of 600 μM H_2_O_2_ for 1 h. Intracellular ROS scavenging activity was measured using CellRox staining. Representative image of increased intensity of CellRox produced by ROS and decreased intensity on pretreating with histochrome. Data are presented as the mean ± SD of three independent experiments. Scale bar = 100 μm, ### *p* < 0.01 versus -H_2_O_2_ -histochrome; * *p* < 0.05; ** *p* < 0.01 versus +H_2_O_2_ -histochrome, *n* = 3. Error bars indicate S.E.M. (**B**) 2’,7’–difluorofluorescin diacetate (H_2_-DFFDA)assay was used to measure cellular ROS production. *** *p* < 0.001 versus -H_2_O_2_ -histochrome; ###, *p* < 0.001 versus +H_2_O_2_ -histochrome, *n* = 3. Error bars indicate S.E.M. (**C**) After pretreatment with histochrome for 24 h, hCPCs were exposed to H_2_O_2_ for 1 h and mitochondrial superoxide production was measured with MitoSOX staining. Representative image of the increased intensity of MitoSOX and decreased intensity on pretreatment with histochrome. Scale bar = 100 μm.

**Figure 3 marinedrugs-17-00368-f003:**
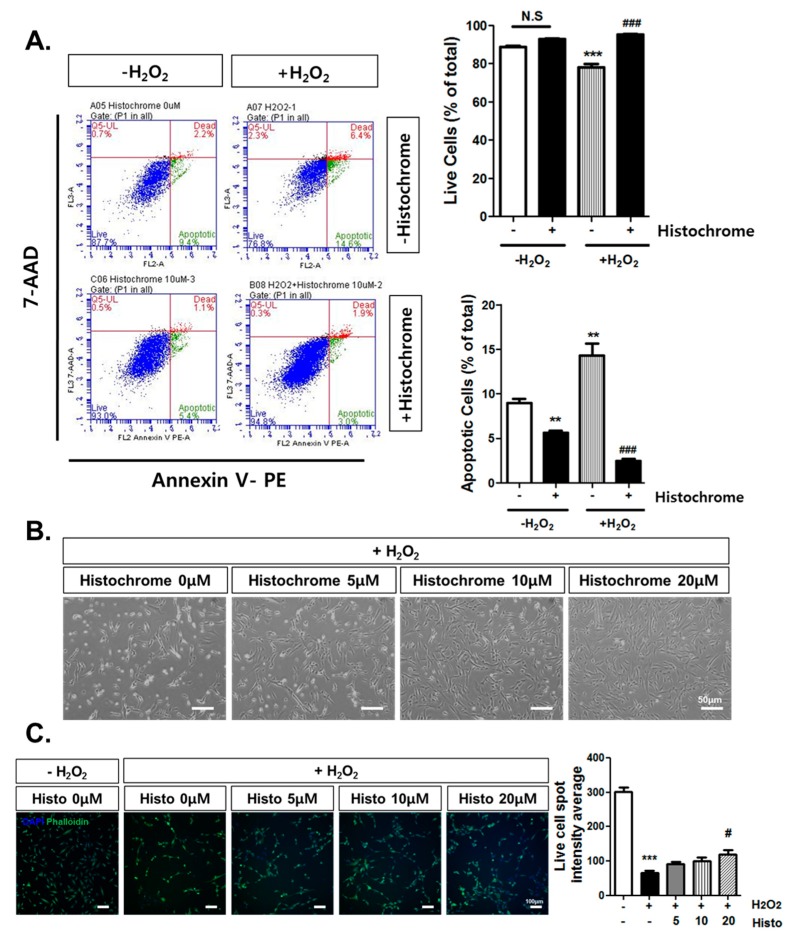
Anti-apoptotic effect of histochrome against H_2_O_2_-induced cell death. (**A**) hCPCs were pretreated with 10 µM of histochrome for 24 h and then exposed to 1 mM H_2_O_2_ for 4 h. Apoptotic cells were quantified by fluorescence-activated cell sorting (FACS) analysis with Annexin V / 7-AAD staining. ** *p* < 0.01; *** *p* < 0.001 versus -H_2_O_2_ -histochrome; ### *p* < 0.001 versus +H_2_O_2_ -histochrome. (**B**) Representative images showing the morphology of hCPCs pretreated with histochrome (0 µM, 5 µM, 10 µM, and 20 µM) in the presence of H_2_O_2_-induced oxidative stress. Morphology of hCPCs was observed by phase contrast microscope. Scale bar = 50 μm (**C**) Live cells were quantified by phalloidin (green fluorescence) intensity. *** *p* < 0.001 versus -H_2_O_2_ -histochrome; # *p* < 0.05 versus +H_2_O_2_ -histochrome. Scale bar = 100 μm, *n* = 3. Error bars indicate S.E.M.

**Figure 4 marinedrugs-17-00368-f004:**
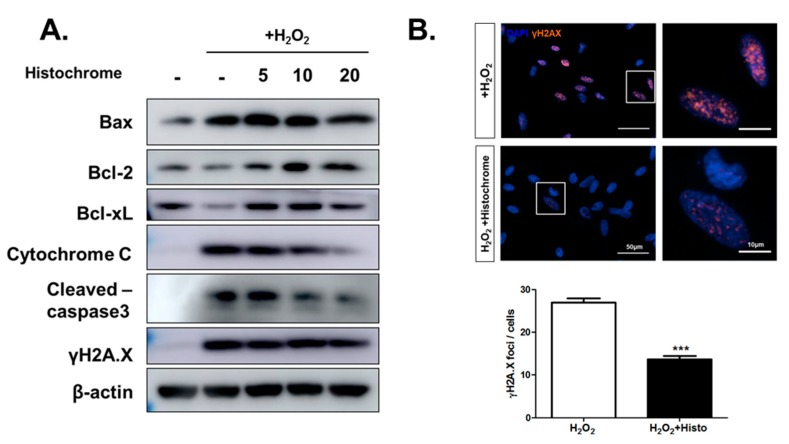
hCPCs pretreated with histochrome show downregulation of pro-apoptotic signals and upregulation of anti-apoptotic signals under the oxidative stress condition. (**A**) hCPCs were pretreated with histochrome for 24 h, oxidative stress was induced in hCPCs by 1 mM H_2_O_2_. Expression of apoptosis signaling-related proteins was determined by western blotting. (**B**) Immunofluorescence was performed with the DNA damage marker γH2A.X to quantify DNA damage of hCPCs. Images were captured using a LionHeart FX automated microscope (Biotek, Winooski, VT, USA). *** *p* < 0.001 versus +H_2_O_2_, *n* = 5. Error bars indicate S.E.M.

**Figure 5 marinedrugs-17-00368-f005:**
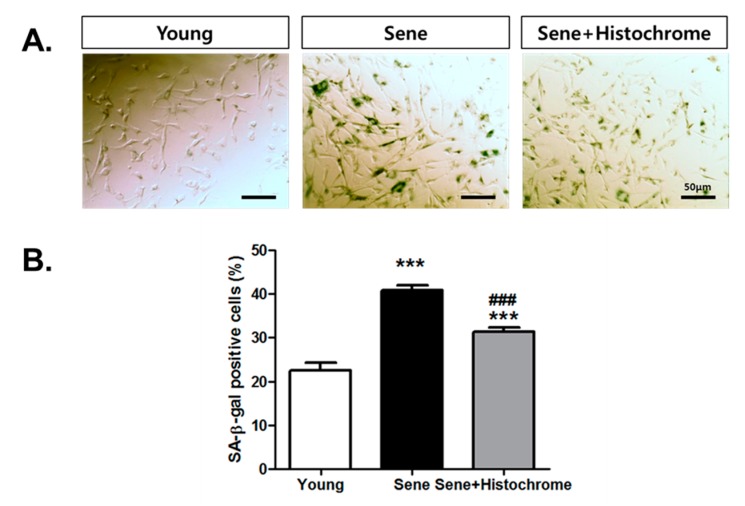
Effect of prolonged treatment with histochrome on hCPCs senescence. (**A**) Representing images of senescence-β-galactosidase (SA- β-gal) stained hCPCs (Scalebar = 50 μm). (**B**) SA- β-gal positive cells were quantified and presented as a graph (*** *p* < 0.001, versus Young; ### *p* < 0.001, versus Sene) Error bars indicate S.E.M. Abbreviation: Sene, senescent hCPCs (passage 13); sene + histochrome, prolonged treatment with histochrome (till passage 13).

**Figure 6 marinedrugs-17-00368-f006:**
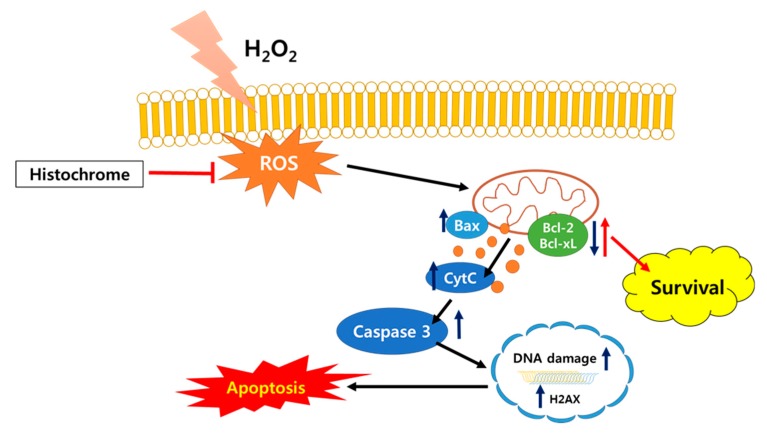
Schematic representation of cytoprotective effects of histochrome against H_2_O_2_-induced cell death via reduction of DNA damage and activation of survival signaling.

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
