# Peer review of "Therapeutic Cell Protective Role of Histochrome under Oxidative Stress in Human Cardiac Progenitor Cells"

_marinedrugs, 2019, doi:10.3390/md17060368_

Round 1
Reviewer 1 Report
The authors have investigated the protective effects of the histochrome on the cardiac progenitor cells. The paper is very clear and has presented the results and discussion in very lucid way. However I do have some comments regarding the paper as follows:
1. The introduction though effective is very brief and succinct. The same goes for the introduction. Further expansion on both these fronts is needed specially from incorporating prior literature point of view to make it a more informative and make the interpretation of results more engaging for the readers in this field.
2. Kindly expand more on the histochrome drug and its use more and its use clinically.
3. Did the authors use any positive control for the cell viability assay not only to validate the assay itself but also to validate the kit. Any previous references of the successful use of the kit in similar setting should be added.
4. The authors should add in the discussion the rationale and the differences of the cell viability assay and death assay and how they contribute different things to the paper
5. How many times were each of the in-vitro experiments conducted and averaged?
6. Furthermore, for cell-based assay like the cell viability assay the authors need to mention total number of repetitions in each treatment group. Were they done in duplicated or triplicates or some other?
7. Scale bars for microscopic images are missing for figure3. Kindly add the scale bars to all the images and report the magnification at which the images were taken.
8. More details on the cell viability kit and assay are needed since no other reference has been added. The authors mention MTS only once on figure 1. This discrepancy needs to be fixed.
9. Better resolution images are needed for the western blots. Also, in figure 3C – the quantification graph is extremely downsized and difficult to read. Kindly restructure the images.
Author Response
Report Form
Open Review
(x) I would not like to sign my review report
( ) I would like to sign my review report
English language and style
( ) Extensive editing of English language and style required
( ) Moderate English changes required
(x) English language and style are fine/minor spell check required
( ) I don't feel qualified to judge about the English language and style
Yes | Can be improved | Must be improved | Not applicable | |
Does the introduction provide sufficient background and include all relevant references? | ( ) | (x) | ( ) | ( ) |
Is the research design appropriate? | ( ) | (x) | ( ) | ( ) |
Are the methods adequately described? | ( ) | (x) | ( ) | ( ) |
Are the results clearly presented? | ( ) | (x) | ( ) | ( ) |
Are the conclusions supported by the results? | (x) | ( ) | ( ) | ( ) |
Comments and Suggestions for Authors
The authors have investigated the protective effects of the histochrome on the cardiac progenitor cells. The paper is very clear and has presented the results and discussion in very lucid way. However I do have some comments regarding the paper as follows:
1. The introduction though effective is very brief and succinct. The same goes for the introduction. Further expansion on both these fronts is needed specially from incorporating prior literature point of view to make it a more informative and make the interpretation of results more engaging for the readers in this field.
->Thank you for your valuable comments. We revised introduction, which is additional information about CPC regeneration potential, ROS effects cellular metabolism and Histochrome clinically application reports. .Additional information described as below. “Recent preclinical studies suggest that transplantation of CPCs into ischemic myocardium significantly improved cardiac regeneration via the formation of vasculature and new cardiomyocyte [8-10]. Furthermore, CPCs had a potential to produce and remodel ECM proteins [11], trigger CPC proliferation and growth factor secretion[12]. According to these positive results, CPC might be one of promising stem cell source in cardiovascular regeneration.”
ROS have been reported that various important development processes, cell signaling, regulation of homeostasis [16]. Low level of ROS involved in regulation of stem cell fate decision, stem cell proliferation, dirferentiation and survival [17].
Echinochrome A is the common sea urchin pigment [20] that has a chemical structure of 6-ethyl-2,3,5,7,8-pentahydroxy-1,4-naphthoquinone (Fig 1A) and exhibits antioxidant, anti-viral [21], anti-inflammatory[22], and anti-diabetic activities[23]. Echinochrome A prevents mitochondrial dysfunction and activation of MAPK cell death signaling pathways caused by cardiotoxic drug treatment [24]. Echinochrome A regulates mitochondria biogenesis in cardiomyocytes by upregulating transcription of mitochondrial regulatory genes such as mitochondrial transcriptional factor A (TFAM), nuclear respiratory factor (NRF-1), and proliferator-activated receptor gamma co-activator (PGC-1α) [25]. Echinochrome A inhibits the phosphorylation of serine-16 and threonine-17, located in the active center of phospholamban, membrane phosphoprotein, the main regulator of the SERCA2A receptor, responsible for the transfer of calcium ions from the cytosol to the sarcoplasmic reticulum preventing ischemic myocardial damage by reducing the infarction zone [26].
2. Kindly expand more on the histochrome drug and its use more and its use clinically.
->Thank you for your valuable comments. In manuscript we mentioned histochrome clinically using information. “Echinochrome A is unsoluble in water, so for medical applications is used its water-soluble sodium salt, which is manufactured under inert conditions in ampoules and is known as the Histochrome® drug. Histochrome has been used in Russia in ophtalmological and cardiological clinical practice. In ophtalmology histochrome is used for the treatment of degenerative diseases of the retina and cornea, macular degeneration; primary open-angle glaucoma; diabetic retinopathy; hemorrhage in the vitreous body, retina, anterior chamber; dyscirculatory disorder in the central artery and vein of the retina[27]. .The overview of clinical applications of histochrome in cardiology is presented in review. In the first place, histochrome has been used for treatment of myocardial ischemia/reperfusion injury. Even a single injection of histochrome immediately after reperfusion recovered the ECG signs of myocardial necrosis and significantly (up to 30%) reduces the necrosis zone after a 10-day course.
The use of histochrome prevented lipid peroxidation, reduced the frequency of left ventricular failure, did not affect the level of blood pressure and heart rate, and decreased the frequency of post-infarction angina pectoris. In drug safety issue, there was no reporting any serious adverse effects for clinical application.
3. Did the authors use any positive control for the cell viability assay not only to validate the assay itself but also to validate the kit. Any previous references of the successful use of the kit in similar setting should be added.
Thank you for your advice. In this study, we using D-PlusTM CCK-3000kit to confirmed cell viability. The product reacts with dehydrogenase in living cells with WST as a substrate to produce soluble formazan. In previous sensitivity test, D-Plus CCK-3000 tetrazolium salts sensitivity was most highest other salts such as WST-1, XTT, MTS. And stability at 4℃ was higher than other MTS based solution.
Previous 2018 Jimin Kim et al. (Oncotarget) revealed that HX-1171 attenuated pancreating b-cell apoptosis and hyperglycemia-mediated oxidative stress via Nfr2 activation using this cell viability kit1. And 2019 Donghee Kim et al reported at the Evidence Based completion and alternative Medicine journal, they used CCK-3000 kit and conducted cell viability assay2.
4. The authors should add in the discussion the rationale and the differences of the cell viability assay and death assay and how they contribute different things to the paper
->Thank you for your kindly comments. In figure 1B revealed that cytotoxicity of histochrome. The cell viability assay performed to check the cell toxicity by difference concentration of histochrome. So we revised the cell viability assay to cytotoxicity assay. Cell death assay was conducted under H2O2 induced oxidative stress. In figure 3, we examined whether histochrome has anti-apoptotic effect. At first, we conducted AnexinV/7-AAD assay which revealed that histochrome has anti-apoptotic effect against oxidative stress. In figure 3C calcein live cell staining showed that double confirm of previous results (increase of live cell). In these result shows that histochrome has antiapoptotic effect against oxidative stress.
5. How many times were each of the in-vitro experiments conducted and averaged?
->Thank you for your kindly comments. All data were obtained from at least 3 independent experiments. We revised how many experiment we did in the figure legend.
6. Furthermore, for cell-based assay like the cell viability assay the authors need to mention total number of repetitions in each treatment group. Were they done in duplicated or triplicates or some other?
->Thank you for your kindly comments. Cell viability assay was conducted six repetitions. We mentioned how many times we did experiments in manuscript.
7. Scale bars for microscopic images are missing for figure3. Kindly add the scale bars to all the images and report the magnification at which the images were taken.
Thank you for your advice. We added scale bars all images as described blow. (Figure 2A, Figure 3B & C)
8. More details on the cell viability kit and assay are needed since no other reference has been added. The authors mention MTS only once on figure 1. This discrepancy needs to be fixed.
->Thank you for your valuable comments. We revised cell viability kit related reference1,2,3 and additional method in Figure legend as described below. “The cell viability was measured by incubating cells with CCK solution. The plates were incubated for 1h and absorbance of the well was measured.” And we added reference information in manuscript.
9. Better resolution images are needed for the western blots. Also, in figure 3C – the quantification graph is extremely downsized and difficult to read. Kindly restructure the images.
-> We changed western blot image. And we also changed the quantification graph size in Figure 3C. Thank you for your valuable comments.
Reference.
1. Jimin Kim et al. HX-1171 attenuates pancreatic β-cell apoptosis and hyperglycemia-mediated oxidative stress via Nrf2 activation in streptozotocin-induced diabetic model. Oncotarget. 2018 9(36): 24260–24271.
2. Jimin Kim et al. Bacterial Clearance Is Enhanced by α2,3- and α2,6-Sialyllactose via Receptor-Mediated Endocytosis and Phagocytosis. Infect Immun. 2019, 87(1): e00694-18.
3. Donghee Kim et al. Protective Effects of Broussonetia kazinoki Siebold Fruit Extract against Palmitate-Induced Lipotoxicity in Mesangial Cells. 2019, 4509403, 12
Reviewer 2 Report
The manuscript by Park et al. is focused in protective role of histochrome under oxidative stress in human cardiac progenitor cells. After close evaluation of paper I suggest minor revision according to next points:
Antioxidant activity can be found in may plant and natural products. It is very important to discuss the results of the study with in vivo effects which were found in vivo.Therefore I suggest to discuss results of study using data published in recent reviews (see doi: 10.1039/C8RA04777D; doi: 10.1007/s11101-018-9547-3)
Histochrome is a sodium salt of Echinochrome A. Please describe in Section 4.1 how this sodium salt was received.
I would suggest to clarify in the abstract as well as in the conclusion that the sodium salt of echinochrome A was used in the study.
Author Response
Open Review
(x) I would not like to sign my review report
( ) I would like to sign my review report
English language and style
( ) Extensive editing of English language and style required
( ) Moderate English changes required
( ) English language and style are fine/minor spell check required
(x) I don't feel qualified to judge about the English language and style
Yes | Can be improved | Must be improved | Not applicable | |
Does the introduction provide sufficient background and include all relevant references? | ( ) | (x) | ( ) | ( ) |
Is the research design appropriate? | (x) | ( ) | ( ) | ( ) |
Are the methods adequately described? | ( ) | (x) | ( ) | ( ) |
Are the results clearly presented? | ( ) | (x) | ( ) | ( ) |
Are the conclusions supported by the results? | (x) | ( ) | ( ) | ( ) |
Comments and Suggestions for Authors
The manuscript by Park et al. is focused in protective role of histochrome under oxidative stress in human cardiac progenitor cells. After close evaluation of paper I suggest minor revision according to next points:
Antioxidant activity can be found in may plant and natural products. It is very important to discuss the results of the study with in vivo effects which were found in vivo. Therefore I suggest to discuss results of study using data published in recent reviews (see doi: 10.1039/C8RA04777D; doi: 10.1007/s11101-018-9547-3).
Thank you for your valuable comment. Of course, we are familiar with those review articles regarding spinochromes distribution and bioactivity. In our manuscript, we tried to refer to original works, especially because mentioned reviews are focused on isolation, distribution of spinochrome pigments and their synthesis, and we need to discuss confirmed clinical effects of echinochrome A. Monography of Afanas’ev contains the whole review of clinical use of histochrome in cardiology.
Histochrome is a sodium salt of Echinochrome A. Please describe in Section 4.1 how this sodium salt was received.
Next experimental paragraph was added:
“The standardized substance echinochrome A (registration number in Russian Federation is P N002362/01) was isolated from the sea urchin Scaphechinus mirabilis by a previously described method [47]. The purity of echinochrome A (99.0%) was confirmed by LC-MS data (Shimadzu LCMS-2020, Kyoto, Japan). Purified echinochrome A appeared like red-brown needles, had a melting point of 221 °С; and similar NMR spectra to that reported previously [47]. We used a solution of echinochrome A sodium salts in ampoules with tradename Histochrome®. Histochrome is obtained by interaction of echinochrome A (1 g) with sodium carbonate (0.4 g) in water solution heated in inert gas until complete removal of CO2. This solution with concentration of echinochrome A 0.2 mg/ml (750 μM) is sealed in ampoules in inert gas. After opening of ampoule, histochrome is used as stock solution that can be diluted with needed solvent or culture media.”
I would suggest to clarify in the abstract as well as in the conclusion that the sodium salt of echinochrome A was used in the study.
Thank you, we added comment “Histochrome (sodium salt of echinochrome A – common sea urchin pigment)…” in both abstract and conclusions sections.
Round 2
Reviewer 1 Report
None. All my concerns on the previous version have been addressed.